# The Molecular Mechanism of Polysaccharides from *Polygonatum cyrtonema* Hua in Improving Hyperuricemia by Regulating Key Targets of Uric Acid Metabolism in Mice

**DOI:** 10.3390/foods14193396

**Published:** 2025-09-30

**Authors:** Shoucheng Pu, Jufang Gong, Meihao Sun, Zunhong Hu, Zhihua Wu

**Affiliations:** 1Jinhua Key Laboratory of Biotechnology on Specialty Economic Plants, College of Life Sciences, Zhejiang Normal University, Jinhua 321004, China; pusc@zjnu.edu.cn (S.P.); sky103@zjnu.cn (J.G.); mhsun@zjnu.cn (M.S.); 2China-Mozambique “Belt and Road” Joint Laboratory on Smart Agriculture, Zhejiang Normal University, Jinhua 321004, China; 3Yunnan Key Laboratory of Genetic Improvement of Herbal Oil Crops, Industrial Crops Research Institute, Yunnan Academy of Agricultural Sciences, Kunming 650225, China; zhhu@yaas.org.cn

**Keywords:** *Polygonatum cyrtonema*, polysaccharides, hyperuricemia, molecular mechanism, xanthine oxidase, URAT1, inflammatory factors

## Abstract

*Polygonatum cyrtonema* Hua, a plant with a long history of consumption in China, serves both medicinal and edible purposes, and it exhibits numerous pharmacological effects, including promoting kidney health and enhancing immune function. However, the effect and molecular mechanism of *Polygonatum cyrtonema* polysaccharides (PCPs) on hyperuricemia have not yet been reported. The hyperuricemic mice model was induced by the intraperitoneal injection of potassium oxonate (PO, 300 mg/kg), combined with the intragastric administration of hypoxanthine (HX, 300 mg/kg). Biochemical assays in mice revealed that PCPs markedly lowered high serum uric acid levels, suppressed xanthine oxidase (XOD) activity, and reduced the expression of inflammatory cytokines, including interleukin-1β (IL-1β), interleukin-6 (IL-6), and tumor necrosis factor-alpha (TNF-α). Western blot analysis demonstrated that PCPs downregulated urate transporter 1 (URAT1), while H&E staining showed that PCPs effectively restored renal histological integrity. Here, we isolated and identified the PCPs, which consist mainly of rhamnose, glucuronic acid, galacturonic acid, glucose, galactose, and arabinose, with a molar mass ratio of 0.5:2.15:0.47:16.58:3.66:1.09. Furthermore, the galactose residue that docked with both XOD and URAT1 molecules forms more hydrogen bonds and exhibits a lower binding energy, which enables the improved regulation of both targets. We have demonstrated for the first time the improving effect of PCPs on hyperuricemia, and revealed their regulatory mechanisms by modulating xanthine oxidase, inflammatory factors, and uric acid transporters. This study not only provides new insights into the anti-hyperuricemic activity of PCPs in mice, but also lays a foundation for its potential application in the functional foods of anti-hyperuricemia.

## 1. Introduction

With the continuous improvement in living standards, high serum uric acid levels have emerged as a significant risk factor for human health. This is largely due to the increased consumption of high-purine meats and flavoring plants, along with irregular daily routines, excessive alcohol intake, and a lack of physical exercise [1,2]. Hyperuricemia (HUA) is a metabolic disorder characterized by disruptions in purine metabolism, resulting in elevated levels of serum uric acid or impairments in uric acid excretion [3,4]. HUA can precipitate chronic kidney disease, metabolic syndrome, diabetes, obesity, hypertension, and cardiovascular or cerebrovascular diseases, posing a significant threat to public health [3,5,6,7]. In the absence of clinical symptoms, the condition is referred to as “the asymptomatic HUA” [8,9]. In severe instances, an elevated concentration of serum uric acid can result in the deposition of urate crystals in joints, kidneys, and other tissues, potentially leading to gout or uric acid nephropathy [3,5,6,10,11,12]. The prevalence of HUA in China has risen annually, showing a significantly higher incidence in men than in women. The age of onset among men is showing a younger trend [2,13,14].

The liver is primarily responsible for the formation of uric acid, with xanthine oxidase (XOD) serving as a crucial enzyme in this process [15,16]. The chemical reactions are shown as follows: Hypoxanthine + H_2_O + O_2_ → Xanthine + H_2_O_2_, Xanthine + H_2_O + O_2_ → Uric Acid + H_2_O_2_. It sequentially oxidizes hypoxanthine to xanthine, and then xanthine to uric acid, while simultaneously producing hydrogen peroxide and reactive oxygen species (ROS), such as superoxide radicals [13,17]. These radicals intensify the immune-inflammatory response, leading to the release of inflammatory factors such as IL-1*β*, IL-6, and TNF-α, which can be detrimental to human health [5,6,13,17]. The kidneys play a major role in the excretion of uric acid, with URAT1 managing UA reabsorption [15,16,17]. Over two-thirds of the body’s uric acid is eliminated through the renal regulation of these transporter proteins [18,19]. Disruptions in purine metabolism, resulting in either excessive uric acid production by the liver or reduced renal excretion, can raise serum uric acid (UA) levels, directly causing HUA.

Typical treatments for HUA involve inhibitors of HUA production such as allopurinol and febuxostat, agents that promote HUA excretion like benzbromarone and probenecid, and medications that aid in dissolving HUA, including rasburicase and pegloticase [17]. Nonetheless, the prolonged use of these medications can lead to a range of side effects, including joint inflammation, allergic reactions, digestive issues, suppression of bone marrow activity, and damage to the liver or kidneys, restricting their use in clinical settings [5,6,13]. Consequently, selecting medicinal and edible plants or mushrooms that have the potential to decrease serum HUA levels by regulating multiple targets could serve as an effective strategy for managing HUA and gout, especially in the dietary treatment of both asymptomatic and chronic HUA [20,21,22,23]. These medicinal and edible plants or mushrooms are well-suited for prolonged use in individuals with chronic conditions due to their complex composition, multiple therapeutic targets, and minimal adverse effects. These functional components, which have a hypouricemic effect by modulating XOD, inflammatory factors, and HUA transporters, and by protecting the kidneys in hyperuricemic mice, include saponins [18], flavonoids [21,24], polyphenols [25], polysaccharides [26], and ingredients like mushrooms [27].

*Polygonatum cyrtonema* Hua (*P. cyrtonema*), which belongs to Asparagaceae, is a traditional herbal medicine, having both medicinal and edible value in its dried rhizome [28]. It is predominantly distributed in the northern, eastern, and southern regions of China, notably within the provinces of Henan, Hubei, Sichuan, and Guizhou. The plant is officially documented in the Chinese Pharmacopeia [29]. Its natural phytochemicals are suitable for long-term consumption and can be utilized in conjunction with chemical pharmaceuticals for supplementary treatment. *P. cyrtonema* was first documented in the “Biography of Famous Physicians” and is also mentioned in ancient texts such as the “Compendium of Materia Medica” [30,31]. The efficacy of traditional Chinese medicine encompasses functions such as promoting spleen and kidney health and improving lung problems, among others. It is effective in treating conditions like spleen deficiency, physical fatigue, lung deficiency with a dry cough, and kidney deficiency with dizziness [30,31,32]. *P. cyrtonema* is supervised by the Ministry of Health and Center for Food Evaluation (State Administration for Market Regulation) in China.

The chemical constituents of *P. cyrtonema* include polysaccharides, steroidal saponins, and flavonoids, among others. Notably, polysaccharides constitute the primary component. *P*. *cyrtonema* polysaccharides (PCPs) demonstrate a variety of pharmacological effects, encompassing antioxidant, anti-aging, and anti-microbial properties, the enhancement of immune function, the regulation of serum glucose and lipid levels, anticancer activities, effects on obesity, the protection of cardiocerebrovascular health, and anti-fatigue properties [31,32,33,34,35,36,37,38,39]. Clinically, *P. cyrtonema* is employed in the treatment of diabetes, diabetic nephropathy, cardiocerebrovascular diseases, pulmonary tuberculosis, male infertility, and gynecological disorders [30,31,33]. PCPs are one of the main bioactive components, and play a significant role in the treatment and prevention of various diseases. However, the effects and mechanisms of PCPs in HUA remain unclear.

Based on the function of PCPs in promoting kidney health and antioxidant activity, and regulating immune activity, this study carried out the separation and purification of PCPs and the identification of structural components. The molecular docking simulation of polysaccharide residues and target protein (XOD and URAT1) molecules was carried out, and the efficacy and mechanisms of PCPs in lowering HUA were explored through animal efficacy experiments. This study establishes the experimental foundation for the development of distinctive health products with the novel function of reducing HUA levels.

## 2. Materials and Methods

### 2.1. Reagents and Materials

1-phenyl-3-methyl-5-pyrazolone (PMP, P109105) was purchased from Shanghai Aladdin Biochemical Co., Ltd. (Shanghai, China). Allopurinol (AP, Cat No:A800424), potassium oxonate (PO, Cat No:P831461), and hypoxanthine (HX, Cat No:H811076) were purchased from Shanghai Macklin Biochemical Co., Ltd. (Shanghai, China). Assay kits for determination of Serum UA (Cat No:105-000476-00), Urea (Cat No:105-000452-00), and creatinine (CREA, Cat No:105-000457-00) were obtained from Shenzhen Mindray Biomedical Electronics Co., Ltd. (Shenzhen, China). Enzyme-linked immunoabsorbent commercial cytokines for murine TNF-α (Cat No: RX202412M), IL-1β (Cat No: RX203063M), and IL-6 (Cat No: RX203049M) were bought from Quanzhou Ruixin Biotechnology Co., Ltd. (Quanzhou, China). Xanthine oxidase (XOD, Cat No: A002-1-1) assay kit was bought from Nanjing JianCheng Bioengineering Institute (Nanjing, China). Anti-URAT1(Cat No:14937-1-AP), anti-β-actin (Cat No:81115-1-RR), and secondary antibody (Cat No:SA00001-2) were purchased from Proteintech Group, Inc. (Wuhan, China), and Nitrocellulose Blotting Membrane (NC, 0.45 µm) was supplied by Amersham (Sheffield, UK). Agilent 1260 HPLC was bought from Agilent Technologies, Inc. (Santa Clara, CA, USA). *Polygonatum cyrtonema* Hua was from the planting base of Jinhua City, Zhejiang Province.

### 2.2. Extraction, Purification, and Content Determination of PCPs

The extraction and purification process of polysaccharides from *P. cyrtonema* is referred to in the literature [28]. A total of 300 g of *P. cyrtonema* slices was crushed, mixed with distilled water at a 1:10 (g/mL) ratio, and sonicated at 40 Hz for 10 min (4 s on, 4 s off) for 10 cycles (a total of 100 min). After ultrasonication, the mixture was filtered, and the residue was re-extracted twice. The third extraction was followed by centrifugation or gauze filtration to avoid clogging. The combined filtrates were stored at 4 °C as the polysaccharide extract.

The extract was concentrated under reduced pressure until turbid, then centrifuged at 10,000 rpm for 5 min at 4 °C to clarify. An equal volume of anhydrous ethanol was added, and the mixture was refrigerated overnight. After filtration and drying, crude polysaccharides were obtained. The crude polysaccharides often contain proteins and were purified using a modified Sevage method. The crude polysaccharides were dissolved in 200 mL water, mixed with a chloroform: n-butanol (5:1) Sevage reagent at a 1:5 ratio, shaken vigorously for 15 min, and centrifuged at 10,000 rpm for 5 min at 4 °C. The supernatant was collected and re-extracted four times. After concentration and ethanol precipitation overnight, purified PCPs were obtained by centrifugation and membrane filtration. Then, the PCP content was determined using the anthrone colorimetric method.

### 2.3. Analysis of PCP Monosaccharide Composition by HPLC

PCPs were subjected to hydrolysis into monosaccharides, which were analyzed by reverse-phase HPLC after PMP derivatization [26,40]. The hydrolysis of polysaccharides was conducted using trifluoroacetic acid (TFA) at a temperature of 100 °C for a period of 6 h. After allowing the mixture to cool to room temperature, the acid was removed by evaporation under a nitrogen stream. Next, a solution of 1-phenyl-3-methyl-5- pyrazolone (PMP) and sodium hydroxide (NaOH) was added to the hydrolyzed sample (or a mixture of standard monosaccharides) and incubated in a water bath at 70 °C for 1 h. Following this, the pH was adjusted to 7, and the product was extracted with chloroform. The chloroform layer was removed, and the residual aqueous layer was subsequently filtered through a 0.45 μm membrane prior to analysis via high-performance liquid chromatography (HPLC).

The analysis of the monosaccharide composition of PCPs was conducted using HPLC with a Waters XB—C18 column (250 mm × 4.6 mm, 5 μm) maintained at 30 °C. The detection wavelength was set at 250 nm. The mobile phase consisted of a gradient elution of acetonitrile and phosphate buffer (pH = 7.0) at a flow rate of 1 mL/min. The mobile phase was an acetonitrile: (pH = 7.0) phosphate buffer = 16%:84% gradient elution at a flow rate of 1 mL/min; the HPLC injection volume was 20 µL.

### 2.4. Animals and Experimental Protocols

A total of 48 male Kunming mice (20 ± 2 g) were supplied by Zhejiang Weitong Lihua Laboratory Animal Technology Co., Ltd, Tongxiang Branch (ZheJiang Jiaxing, China) (Permit ID: 20230525Abzz0600999711). Mice were kept in standard conditions under a 12 h light/dark cycle in a controlled room with a temperature of 25 ± 1 °C and 50 ± 10% humidity. All animals were allowed free access to laboratory diet and water for 7 days before performing any procedure.

The mice were randomly divided into six groups, with each group consisting of eight mice: the normal control group (NC), the model control group (MC), the positive control group receiving allopurinol at a dosage of 5 mg/kg/d BW (AP, 0.5 mg/mL), the PCPs–low-dose group administered 125 mg/kg/d BW (LD), the PCPs–medium-dose group given 250 mg/kg/d BW (MD), and the PCPs–high-dose group treated with 500 mg/kg/d BW (HD).

According to a previous study and Table 1 [41], the hyperuricemic mice model was induced by the intraperitoneal injection of potassium oxonate (PO, 300 mg/kg), combined with intragastric administration of hypoxanthine (HX, 300 mg/kg) 1 h before the drug interventions for all groups except the NC group for 14 consecutive days at 8:30 a.m. The NC group was administered with the same volume of 0.5% CMC-Na without HX via intragastric administration and 0.5% CMC-Na without PO via intraperitoneal injection. One hour after modeling, mice in the AP group were intragastrically given allopurinol (5 mg/kg) and mice in the PCP groups intragastrically received PCPs (125, 250, and 500 mg/kg) once daily for 14 days, respectively. All animals were anesthetized one hour following drug administration. Blood samples were collected from the orbital sinus and subjected to centrifugation at 3000 rpm for 10 min at 4 °C. Subsequently to blood collection, the liver and kidneys were excised, rinsed, and promptly weighed. Liver tissue was sectioned, combined with a 0.9% physiological saline solution at approximately nine times the sample weight, and homogenized on ice. The resulting homogenate was centrifuged at 12,000 rpm at 4 °C for 10 min, and the supernatants were collected and frozen for subsequent XOD activity assay. Kidney tissues were preserved for histopathological examination and Western blot analysis, respectively.

### 2.5. Biochemical Assays

Serum concentrations of uric acid (UA), creatinine (CREA), and urea were quantified using a biochemical analyzer. Xanthine oxidase (XOD) activity in the supernatants of liver homogenates was assessed utilizing standard kits in accordance with the manufacturers’ protocols. Protein content in the liver was determined through a bicinchoninic acid (BCA) protein assay kit, using bovine serum albumin as a reference standard. The levels of TNF-α, IL-1β, and IL-6 in serum were measured using an enzyme-linked immunosorbent assay (ELISA) following the instructions provided with the kits.

### 2.6. Histological Assay

Kidney tissues were treated with a 4% paraformaldehyde solution to facilitate paraffin embedding, sectioning, and subsequent haematoxylin and eosin (H&E) staining. Based on H&E staining, histological alterations such as glomerulus and renal tubules were evaluated under a digital scanner from 3DHISTECH (Budapest, Hungary) [42]. (Wuhan Servicebio Technology Co., Ltd., in China for renal HE staining and histopathological examination).

### 2.7. Western Blot Analysis

The kidneys were washed with cold PBS two to three times and subsequently homogenized in nine equivalent volumes of ice-cold RIPA lysis buffer, with the protease inhibitor added shortly before use. After being left on ice for 30 min, the homogenate was centrifuged at 12,000 rpm for 10 min at 4 °C to obtain total proteins, and the concentration was determined using the BCA method. The samples were then subjected to SDS-polyacrylamide gel electrophoresis and transferred onto NC membranes. The membranes were blocked with TBST containing 5% skim milk for 2 h and incubated with the following primary antibodies at 4 °C overnight: anti-URAT1 (1:3000) and anti-β-Actin (1:20,000). This was followed by incubation with a secondary HRP-conjugated goat anti-rabbit IgG (Immunoglobulin G, 1:10,000) antibody at room temperature for 2 h. Finally, the membranes were treated with ECL and photographed using a chemiluminescence imager. The protein bands were analyzed using ImageJ 1.54p software and normalized to the expression of β-Actin.

### 2.8. Molecular Docking Simulation

The 3D structures of receptor proteins were downloaded from the PDB website, and those of ligand molecules were obtained from Pubchem. Ligand molecules included Galactose (Compound CID: 6036), Galacturonic acid (Compound CID: 439215), Glucuronic acid (Compound CID: 94715), and Arabinose (Compound CID: 43919). Using Autodock Vina, we performed molecular docking simulations between receptor proteins (like XOD ID:1FIQ and URAT1 ID:9JDV) and ligands (such as polysaccharide residues) to screen for binding interactions and to calculate binding modes and affinities. We adopted a docking method where the protein is rigid, while the ligand can rotate. The location of the receptor protein binding center is determined based on the ligand position from the PDB website. There is a fixed docking center location for URAT1 (center_x = 82, center_y = 94, center_z = 91) and XOD (center_x = 21, center_y = 16, center_z = 104). The scoring function utilized by AutoDock Vina was set to default. Visual analyses were performed by open-source PyMOL2.4.0 and Discovery Studio visualizer 2025.

### 2.9. Statistical Analysis

Statistical analyses were conducted using SPSS version 19.0. The data are expressed as mean ± standard deviation (SD). Comparisons between two groups were performed using a Student’s *t*-test, with *p* < 0.05 considered indicative of statistical significance.

## 3. Results

### 3.1. Preparation and Monosaccharide Composition Analysis of PCPs

A standard curve (y = 0.0989x + 0.0001) was obtained by reacting glucose solutions of known concentrations with an anthrone reagent. For polysaccharide content determination, 0.1 mL of the sample was mixed with 0.9 mL of water and 3 mL of the anthrone reagent, followed by boiling and spectrophotometric measurement.

The standard curve (y = 0.0989x + 0.0001) for quantitative analysis was obtained by reacting glucose solutions of known concentrations with an anthrone reagent. After the extraction and purification of PCPs, the content of polysaccharides in the sample was calculated based on the standard curve equation (Figure 1a). The retention times of the monosaccharides were determined from the standard samples, and the composition of the polysaccharide samples was determined by comparing the retention times and peak areas with those of the standards. According to the HPLC chromatograms of monosaccharides from standard samples and polysaccharide samples, we found that PCPs were composed of rhamnose, glucuronic acid, galacturonic acid, glucose, galactose, and arabinose, with a molar mass ratio of 0.5:2.15:0.47:16.58:3.66:1.09 (Figure 1b). The results showed that the polysaccharide samples contained a complex mixture of monosaccharides, with varying proportions of each component. Our HPLC analysis of PCPs provided a detailed profile of the monosaccharide composition in *P. cyrtonema* slices, which is essential for subsequent research on their structure–activity relationships.

### 3.2. PCPs Decreased the Level of Serum UA, CREA, and Urea in Mice

After the intraperitoneal injection of potassium oxonate (PO) and intragastrical administration of hypoxanthine for 14 days, the level of serum uric acid in the model group was significantly higher than that of the normal control group (NC) (*p* < 0.01) (Figure 2), indicating that the model was successful in inducing hyperuricemia in mice. The levels of serum uric acid in the PCP treatment groups at doses of 125 mg·kg^−1^ (LD), 250 mg·kg^−1^ (MD), and 500 mg·kg^−1^ (HD) and the allopurinol treatment group (AP) were significantly lower than that of the model group (*p* < 0.01), with no significant difference compared to the normal group.

A remarkable difference was observed between the normal control group and AP group in the levels of serum CREA and urea (*p* < 0.05 or *p* < 0.01). Consistent hyperuricemia or allopurinol might induce kidney damage, leading to the increased serum urea levels in the model and AP groups (*p* < 0.05 or *p* < 0.01) compared with the normal group. However, MD and HD remarkably reduced the serum urea level (*p* < 0.05) compared with the model group, indicating a potential nephroprotective effect of PCPs [42].

### 3.3. PCPs Effectively Inhibited Hepatic XOD Activity in Mice

The liver serves as the primary organ responsible for uric acid production. By reducing the activity of XOD in the liver, the production of uric acid can be effectively decreased. The results showed that PCPs remarkably inhibited hepatic XOD activities compared with the model group (*p* < 0.05), similar to the AP group (Figure 3). XOD activity showed a decreasing trend from low- to high-dose groups of PCPs. There was no significant difference in XOD activity between each PCP dose group and the normal group. These indicate that PCPs restore XOD activity in hyperuricemic mice to levels similar to those in the normal group.

### 3.4. PCPs Attenuated the Increased Inflammatory Cytokines Production in Mice

The levels of inflammatory cytokines TNF-α, IL-1β, and IL-6 in serum were determined (Figure 4). The levels of TNF-α, IL-1β, and IL-6 in PO/HX-induced hyperuricemic mice and in AP group mice notably increased compared with the control group (*p* < 0.05 or *p* < 0.01). Furthermore, 250 and 500 mg/kg PCPs remarkably attenuated an increased inflammatory cytokine production relative to the model group. No significant difference was observed between the PCP treatment groups and the normal control group. These findings suggested that PCPs could effectively ameliorate PO/HX-induced inflammation.

### 3.5. Renal Histopathological Features

The renal units perform essential functions, such as glomerular filtration and the selective reabsorption and secretion processes in the tubules. The results of the pathological section examinations of the kidneys of mice in each group are shown (Figure 5). In the figure, the black arrow indicates the glomerulus, the green arrow points to the renal capsule, the blue arrow highlights the proximal convoluted tubule, and the red arrow marks the distal convoluted tubule. The kidneys of mice in the normal group (Figure 5a) had a normal tissue morphology and structure, normal glomerular structure, and normal tubular epithelial cell structure [43,44]. The gaps between renal tubules were obvious, and the lumen was obvious.

In the HUA model group (Figure 5b), the glomerular structure was loose and disorganized, the cells were swollen, and some of the renal capsule structure changed from a round shape to an irregular shape. The glomerular capsule lumen was reduced or even disappeared, and in some places the capsule was adherent, the renal tubular epithelial cells were mostly swollen and necrotic, the renal tubules were dilated, the tubular lumen was enlarged, some of the renal tubular epithelial cells necrolyzed and detached into the tubular lumen, the internal brush border structure disappeared, and the vacuolar degeneration was severe.

The kidneys of mice in the positive allopurinol group (Figure 5c) had some adhesions of the glomerulus and renal capsule. The epithelial cells of renal tubules were mostly swollen and necrotic, and showed vacuolar degeneration, with a severe luminal dilatation and a partial blockage of the renal tubules. The renal parenchyma exhibited numerous interstitial spaces, and brownish-black infiltrated inflammatory cells were observed within the renal tubules and interstitium. This evidence suggests significant renal injury.

Compared with the model group, the tubular lumen dilatation of renal tubules in mice in the group of low-dose PCPs (Figure 5d) was improved, and the amount of vacuolar degeneration was reduced. Compared with the model group, the tubular lumen of mice in the group of low-dose PCPs improved, the amount of vacuolar degeneration decreased, and the lumen of some renal tubules was clearly visible, and the nuclei of the renal tubular cells in the group of medium-dose PCPs (Figure 5e) were clear, the swelling of the cells was reduced, the gap between the tubules was clearer, and the structure of the glomerulus and the capsule tended to be normalized. By comparison, it can be found that the high-dose PCPs (Figure 5f) had the most obvious protective effect on the renal tissue and structure of the HUA mice.

### 3.6. PCPs Downregulated Renal URAT1 Expression in Mice

PO injection and intragastrical hypoxanthine significantly increased the URAT1 expression levels in kidneys, which could be recovered by PCPs (Figure 6). PCPs (125 mg/kg, 250 mg/kg, and 500 mg/kg) notably decreased URAT1 expression (*p* < 0.05 or *p* < 0.01) in hyperuricemic mice, indicating an inhibitory effect of PCPs on URAT1. AP also restored URAT1 expression abnormalities compared to the model mice (*p* < 0.01).

### 3.7. Molecular Docking Simulation

The molecular docking analysis was employed to elucidate the interaction of PCPs with urate transporters, which are integral to anti-hyperuricemic activity (Figure 7 and Figure 8).

The docking results indicate that the lowest binding energies with XOD for the residues galactose, galacturonic acid, glucuronic acid, and arabinose are −5.9, −6.5, −6.4, and −5.5 kcal/mol, respectively. Similarly, the docking results for URAT1, with the same residues, show binding energies of −4.5, −4.8, −4.8, and −4.2 kcal/mol, respectively.

The interactions (Figure 8a–d) between the polysaccharide and amino acid residues of XOD are primarily facilitated by hydrogen bonding, which effectively inhibits the enzyme’s activity. The galactose residue forms three hydrogen bonds with amino acid residues GLN767, GLN1194, and MET1038, the galacturonic acid residue forms two hydrogen bonds with amino acid residue GLN1194, the glucuronic acid residue forms four hydrogen bonds with amino acid residues ALA1079, SER1080, and GLN1040, and the arabinose residue forms two hydrogen bonds with amino acid residues ALA1079 and SER1080.

The polysaccharide residues inhibit the protein expression of URAT1 mainly through H-bonding interactions (Figure 8e–h), with galactose residues forming four hydrogen bonds with amino acid residues ASN237 and MET214, galacturonic acid residues forming three hydrogen bonds with amino acid residue ASN237, and glucuronic acid residues forming two hydrogen bonds with amino acid residues ASN237 and SER238.

Taking the above comparisons together, galactose residue docked with both XOD and URAT1 molecules is docked with more H-bonds, and the binding energy is low, allowing for a better regulation of the two targets.

## 4. Discussion

Hyperuricemia is a metabolic disorder caused by uric acid metabolism imbalance, leading to elevated serum uric acid levels. While chemical drugs effectively lower the content of uric acid in the human body, they can also damage multiple systems and organs. Recently, polysaccharides from natural sources have become a research focus due to their low toxicity and natural origin [45,46,47].

Polysaccharides, which are high-molecular-weight biopolymers formed through glycosidic bonds between monosaccharides, are prevalent in animals, plants, and microorganisms [48,49]. They offer advantages such as natural bioactivity, low toxicity, good biocompatibility, and easy degradability, ensuring their safety in human applications [48,50]. Their diverse biological activities are attributed to their structure and conformations [51]. The diversity of polysaccharides is determined by factors like monosaccharide types and functional groups [51,52,53,54,55]. For example, carboxyl, sulfonic, and uronosyl groups are involved in antioxidant mechanisms through interactions with target proteins [55,56]. Polysaccharides with a high uronic acid content are more effective in lowering uric acid [44,57]. These structural characteristics facilitate polysaccharides in engaging with target proteins through hydrogen bonds and other bonds, thereby eliciting multidimensional pharmacological effects, including uric acid regulation, antioxidation, anti-inflammation, immune modulation, and hepatoprotection [45,46].

The mechanisms by which plant polysaccharides facilitate the reduction of uric acid are detailed as follows: (1) XOD activity is inhibited to reduce uric acid synthesis [42,45]. (2) The expression of uric acid transporters, like URAT1 and others, is regulated in the kidneys to enhance uric acid excretion [43,45]. (3) Antioxidative effects are exerted by scavenging oxidative stress markers such as ROS and MDA, thereby alleviating organ damage secondary to HUA [58,59]. (4) The immune response is modulated through the inhibition of inflammasome activation and downregulation of pro-inflammatory cytokines like IL-1β and TNF-α [26,45]. (5) The gut microbiota is modulated [43]. Researchers have found that polysaccharides from *Lonicera japonica* significantly decreased the concentration of serum uric acid in rats and inhibited gouty arthritis inflammation by suppressing XOD activity, downregulating URAT1, and exerting anti-inflammatory effects (reducing IL-1β, IL-6, TNF-α, COX-2 levels) by the NF-κB pathway [26]. Natural polysaccharides are hydrolyzed by the digestive system and intestinal microbiota, resulting in the formation of oligosaccharides with varied structural characteristics [45,60]. These oligosaccharides, with a low molecular weight and high solubility, are more efficiently absorbed by cells, enabling the modulation of multiple targets implicated in the management of HUA.

As an edible and medicinal plant, the polysaccharides from *P. cyrtonema* exhibit a good safety [31]. This study examines PCPs, which are composed of rhamnose, glucuronic acid, galacturonic acid, glucose, galactose, and arabinose in a molar mass ratio of 0.5:2.15:0.47:16.58:3.66:1.09. Monosaccharides such as glucuronic acid, galacturonic acid, galactose, and arabinose are notably influential in reducing uric acid levels by modulating antioxidant and immune-inflammatory pathways [45,46]. Galactose, often referred to as “brain sugar,” is essential for cellular formation, immune system functionality, and the facilitation of anti-inflammatory responses [45]. The molecular docking of galactose to its targets through hydrogen bonding has further demonstrated its capacity to modulate the targets XOD and URAT1. Animal studies have demonstrated the efficacy of reducing uric acid levels. Compared to the model group, all dosage groups of PCPs (125/250/500 mg/kg) significantly decreased serum uric acid levels by 21.22–26.47% (*p* < 0.01). The effect observed in the high-dose group of PCPs was comparable to that of the positive control drug, allopurinol, which is approved by the U.S. FDA for the treatment of HUA mice (*p* > 0.05). Notably, high- and medium-dose PCP groups, which had significantly lowered serum urea levels, also had decreased CREA levels and showed less nephrotoxicity than the allopurinol group. The conversion of purine nucleotides into UA involves a highly intricate biochemical pathway. Studies have highlighted the role of XOD as a key enzyme in the biosynthesis of uric acid, where it catalyzes the conversion of xanthine and hypoxanthine into uric acid. All three dosage groups of polysaccharides demonstrated an inhibitory effect on XOD activity in the liver, with the high dose of PCPs exhibiting the most pronounced effect, comparable to the reduction of UA. XOD is inhibited by PCPs, with the low-dose group showing an XOD activity comparable to the normal group, and the high-dose group exhibiting a 30% lower activity than the model group, indicating the potential of PCPs as a natural XOD inhibitor. Histopathological analysis revealed that, while allopurinol effectively lowered the content of uric acid, it caused severe kidney damage and inflammatory cell infiltration, whereas high-dose PCPs best preserved the renal tubular structure, suggesting a kidney-protective effect. The primary method for eliminating UA is through renal excretion. Uric acid transporter 1 (URAT1), encoded by the *SLC22A12* gene, is a crucial component of the urate-anion transporter family, playing a significant role in the reabsorption of more than 90% of uric acid from the lumen into the cells of the kidney’s proximal tubule [42]. The inhibition of URAT1 disrupts the reabsorption process of the urate anion, thereby facilitating an increase in the renal excretion of uric acid [45,46]. PCPs also reduced URAT1 protein expression by 51.5% compared to the model group, thereby boosting uric acid excretion. The present study confirms that PCPs downregulated inflammatory factors, including IL-1β, IL-6, and TNF-α, potentially through the modulation of the NF-κB signaling pathway. PCPs demonstrate efficacy in lowering uric acid levels by inhibiting XOD activity, downregulating urate transporter 1 (URAT1), and exhibiting anti-inflammatory effects, and polysaccharides from *Lonicera japonica* have a similar efficacy. Molecular docking studies reveal that PCPs, including galactose, galacturonic acid, and glucuronic acid, demonstrate lower binding energies with receptor proteins XOD and URAT1, primarily through hydrogen bonding interactions to regulate targets. In contrast, the interaction between arabinose and the URAT1 receptor is devoid of hydrogen bonds, indicating a relatively weaker binding affinity.

## 5. Conclusions

The present study demonstrates that PCPs firstly reduce serum uric acid in mice with HUA. Specifically, the uric-acid-lowering mechanism involves the inhibition of XOD activity, regulation of uric acid transporter proteins, and modulation of inflammatory factors, which collectively contribute to reduced serum uric acid levels in hyperuricemic mice. Molecular docking studies suggest that the residues of PCPs play a role in modulating two critical targets, XOD and URAT1, through hydrogen bonding interactions. The medicinal and edible plant Huangjing (*P. cyrtonema*), due to its unique advantages of safety, long-term edibility, and drug–excipient unity, can be used alone or in combination with chemicals, and will play an active and healthy role in human health care and the treatment of HUA. This study is subject to several limitations, primarily due to the reliance on a mouse model without subsequent validations in rat models and human clinical trials. Subsequent studies need to establish an accurate human quantitative and efficacy relationship model through clinical randomized controlled trials, which will provide an evidence-based basis for the development of health care functional foods.

## Figures and Tables

**Figure 1 foods-14-03396-f001:**
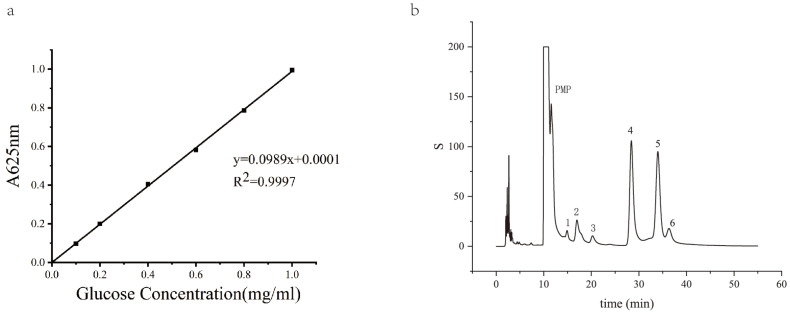
HPLC chromatograms of monosaccharides from standard samples and polysaccharide samples. (**a**) A glucose standard curve for polysaccharide concentration quantification in *P. cyrtonema*. (**b**) HPLC chromatograms of monosaccharides from PCPs.

**Figure 2 foods-14-03396-f002:**
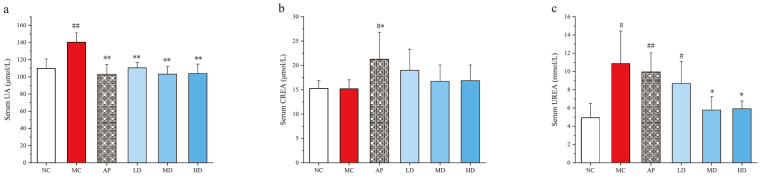
Effects of PCPs on the levels of (**a**) serum UA, (**b**) serum CREA, (**c**) serum urea in hyperuricemic mice. Values are expressed as the mean ± SD (*n* = 8). # *p* < 0.05; ## *p* < 0.01 vs. the NC group; * *p* < 0.05; ** *p* < 0.01 vs. the model group; n.s., no significance. Normal control group (NC), model control group (MC), allopurinol (AP), PCPs–low-dose group (LD), PCPs–medium-dose group (MD), and PCPs–high-dose group (HD).

**Figure 3 foods-14-03396-f003:**
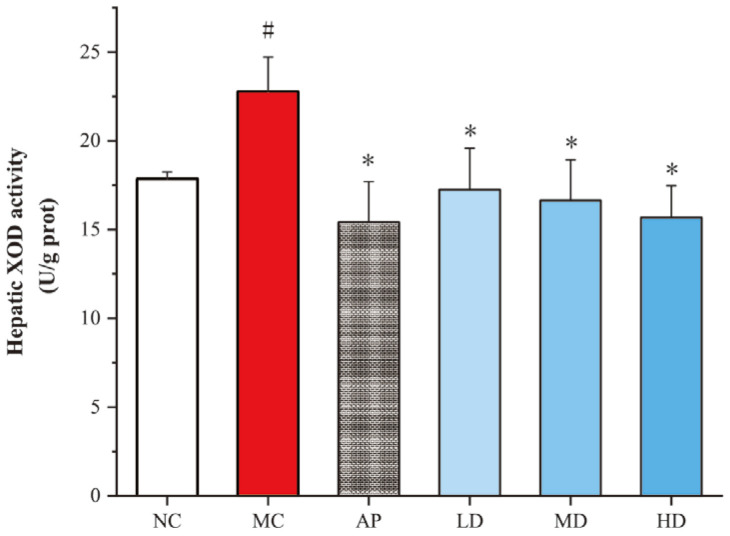
Effects of PCPs on hepatic XOD activity in hyperuricemic mice. Data are presented as the mean ± SD, (*n* = 8). # *p* < 0.05 vs. the NC group; * *p* < 0.05 vs. the model group. Normal control group (NC), model control group (MC), allopurinol (AP), PCPs–low-dose group (LD), PCPs–medium-dose group (MD), and PCPs–high-dose group (HD).

**Figure 4 foods-14-03396-f004:**
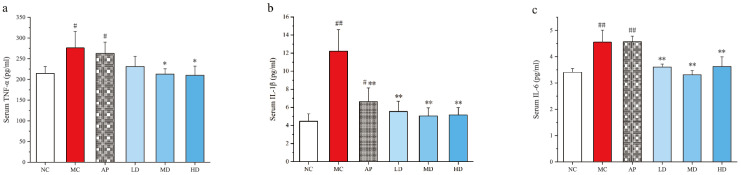
Effects of PCPs on the levels of (**a**) serum TNF-α, (**b**) serum IL-1β, and (**c**) serum IL-6 in PO/HX-induced HUA mice. Values are expressed as the mean ± SD (*n* = 8). # *p* < 0.05; ## *p* < 0.01 vs. the NC group; * *p* < 0.05; ** *p* < 0.01 vs. the model group; n.s, no significance. Normal control group (NC), model control group (MC), allopurinol (AP), PCPs–low-dose group (LD), PCPs–medium-dose group (MD), and PCPs–high-dose group (HD).

**Figure 5 foods-14-03396-f005:**
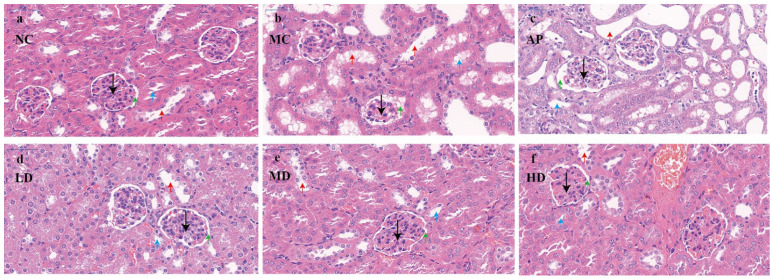
Micrograph of renal histopathology (black arrow: glomerulus; green arrow: renal capsule; blue arrow: proximal convoluted tubule; red arrow: distal convoluted tubule). Normal control group (NC), model control group (MC), allopurinol (AP), PCPs–low-dose group (LD), PCPs–medium-dose group (MD), and PCPs–high-dose group (HD).

**Figure 6 foods-14-03396-f006:**
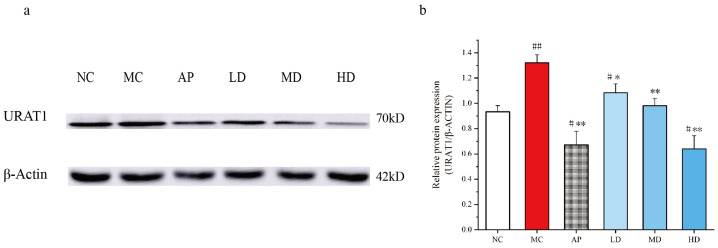
Effects of PCPs on the expression of URAT1 in PO/HX-induced HUA mice. Data are expressed as the mean ± SD, (*n* = 3). # *p* < 0.05; ## *p* < 0.01 vs. the NC group; * *p* < 0.05; ** *p* < 0.01 vs. the model group. Normal control group (NC), model control group (MC), allopurinol (AP), PCPs–low-dose group (LD), PCPs–medium-dose group (MD), and PCPs–high-dose group (HD).

**Figure 7 foods-14-03396-f007:**
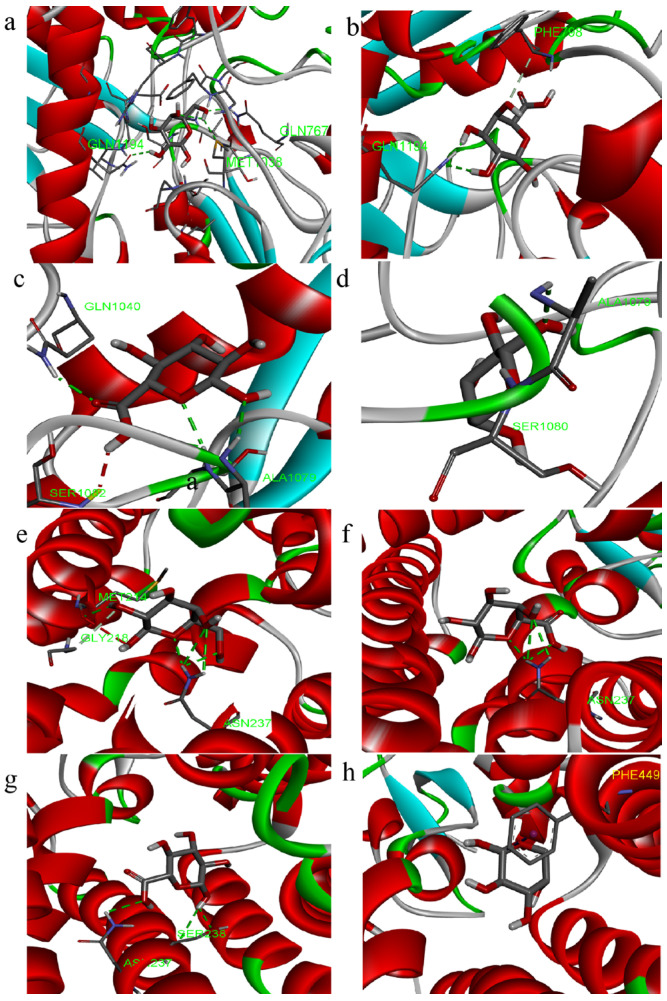
Three-dimensional images of molecular docking ((**a**–**d**): XOD, (**e**–**h**): URAT1; residues: galactose, galacturonic acid, glucuronic acid, and arabinose).

**Figure 8 foods-14-03396-f008:**
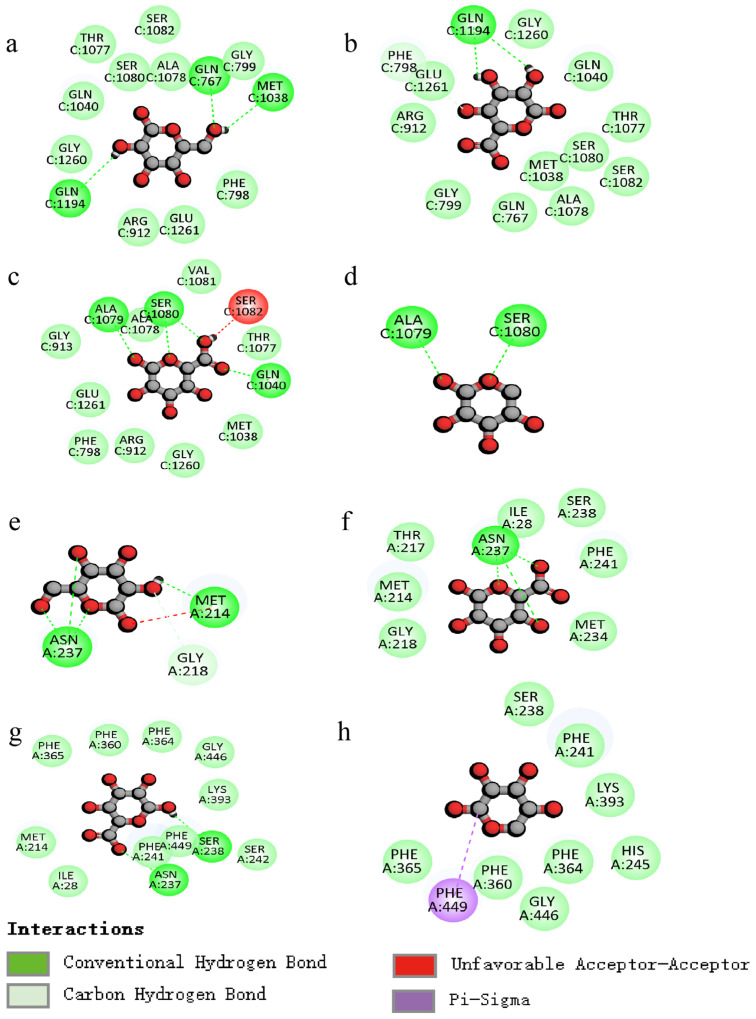
Two-dimensional images of molecular docking ((**a**–**d**): XOD, (**e**–**h**): URAT1; residues: galactose, galacturonic acid, glucuronic acid, and arabinose).

**Table 1 foods-14-03396-t001:** The dosing of each group in mice.

Group	Treatment	
NC	0.5% CMC-Na + 0.5% CMC-Na	0.5% CMC-Na
MC	PO (300 mg/kg/d) + HX (300 mg/kg/d)	0.5% CMC-Na
AP	PO (300 mg/kg/d) + HX (300 mg/kg/d)	AP (5 mg/kg/d)
LD	PO (300 mg/kg/d) + HX (300 mg/kg/d)	PCPs (125 mg/kg/d)
MD	PO (300 mg/kg/d) + HX (300 mg/kg/d)	PCPs (250 mg/kg/d)
HD	PO (300 mg/kg/d) + HX (300 mg/kg/d)	PCPs (500 mg/kg/d)

Note: Normal control group (NC), model control group (MC), allopurinol (AP), PCPs–low-dose group (LD), PCPs–medium-dose group (MD), and PCPs–high-dose group (HD).

## Data Availability

The original contributions presented in this study are included in the article. Further inquiries can be directed to the corresponding author.

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
