# Peer review of "The Molecular Mechanism of Polysaccharides from Polygonatum cyrtonema Hua in Improving Hyperuricemia by Regulating Key Targets of Uric Acid Metabolism in Mice"

_foods, 2025, doi:10.3390/foods14193396_

Round 1
Reviewer 1 Report
Comments and Suggestions for Authors
The submitted work is of interest in mitigating effects of hyperuricemia through dietary intervention. Few comments to be addressed are given below.
Abstract- separate animal studies and invitro results from insilico ones to aid in understanding weight of evidence.
LINES 90-91- Non standardized terms such as moistening lungs used.
Genus species in italics- line 95,101..etc
Was hydrolysis of polysaccahrides optimized? How was complete hydrolysis ensured?
What about non reducing sugar analysis by HPLC/pmp
Figure 3 is with significant error bars. Are the differences between treatments on hepatic XOD activity significant?
How have the author concluded the observed bioactivity is only via polysaccharides (PCPs). Were the extracts characterised for minor components to rule out alternate mode of action.
Comment on global regulatory requirements for such extracts and include in manuscript.
Author Response
Dear editor and reviewers,
Thanks for your critical comments. According to your suggestions, we made revisions and response to reviewers one by one as follows. The comments were listed one by one in numerical order, followed by corresponding answers. The revisions in manuscript were highlighted in red.
Reviewer #1
Comments 1:[ Abstract- separate animal studies and invitro results from insilico ones to aid in understanding weight of evidence.]
Response 1: Thanks. According to your suggestion, we have rewritten the content of abstract as follows (line13-34): Polygonatum cyrtonema Hua, a plant with a long history of consumption in China, serves both medicinal and edible purposes, and it exhibits numerous pharmacological effects, including promoting kidney health and enhancing immune function. However, the effect and molecular mechanism of Polygonatum cyrtonema polysaccharides (PCPs) on hyperuricemia have not been reported yet. The hyperuricemic mice model was induced by intraperitoneal injection of potassium oxonate (PO, 300 mg/kg) combined with intragastric administration of hypoxanthine (HX, 300 mg/kg). Biochemical assays in mice revealed that PCPs markedly lowered high serum uric acid levels, suppressed xanthine oxidase (XOD) activity, and reduced the expression of inflammatory cytokines-including interleukin-1β(IL-1β), interleukin-6 (IL-6), and tumor necrosis factor-alpha (TNF-α). Western blot analysis demonstrated that PCPs down-regulated urate transporter 1 (URAT1), while H&E staining showed that PCPs effectively restored renal histological integrity. Here, we isolated and identified the PCPs, which consist mainly of rhamnose, glucuronic acid, galacturonic acid, glucose, galactose, and arabinose, with a molar mass ratio of 0.5:2.15:0.47:16.58:3.66:1.09. Further, the galactose residue docked with both XOD and URAT1 molecules forms more hydrogen bonds and exhibits lower binding energy, which enables improved regulation of both targets. We have demonstrated for the first time the improving effect of PCPs on hyperuricemia, and revealed their regulatory mechanisms by modulating xanthine oxidase, inflammatory factors and uric acid transporters. This study would not only provide the new insights into the anti-hyperuricemic activity of PCPs in mice, but also lay a foundation for its potential application in functional foods of anti-hyperuricemia.
Comments 2:[LINES 90-91- Non standardized terms such as moistening lungs used.]
Response 2: Thanks for your suggestion. We replaced the sentences with “The efficacy of traditional Chinese medicine encompasses functions such as promoting spleen and kidney health, and improving lung problem, among others.” in corresponding lines 92-93.
Comments 3:[Genus species in italics- line 95,101..etc]
Response 3: Thanks for your suggestion. We replaced the sentences with “Polygonatum cyrtonema Hua (P. cyrtonema)”.
Comments 4:[Was hydrolysis of polysaccahrides optimized? How was complete hydrolysis ensured?]
Response 4: Thanks for your suggestion. We referred to the literature, optimized the hydrolysis time, and carried out derivatization after hydrolyzing for 3, 4, 5, and 6 hours respectively. The monosaccharide content of the samples was stable after 5 hours of hydrolysis. To achieve more thorough hydrolysis, we chose a final hydrolysis time of 6 hours for measurement of monosaccharide.
Comments 5:[What about non reducing sugar analysis by HPLC/pmp]
Response 5: Thanks for your suggestion. we don’t detect non educing sugar.
Comments 6:[Figure 3 is with significant error bars. Are the differences between treatments on hepatic XOD activity significant?]
Response 6: Thanks for your suggestion. There were no significant differences in hepatic XOD activity among the three PCP-treated groups of mice.
Comments 7:[How have the author concluded the observed bioactivity is only via polysaccharides (PCPs). Were the extracts characterised for minor components to rule out alternate mode of action.]
Response 7: Thanks for your suggestion. We used membrane filtration to remove small molecules with a molecular weight lower than 10,00 Da, ruling out alternate mode of action.
Comments 8:[Comment on global regulatory requirements for such extracts and include in manuscript.]
Response 8: Thanks for your suggestion. The Ministry of Health of China and Center for Food Evaluation, State Administration for Market Regulation (CFE) issued a notice on food and medicine (also called Huangjing) being of the same origin. Document No. 36 of 2021 was issued by the Ministry of Health of China, and Document No. 51 of 2002 was issued by the Health and Law Supervision Department.
Moreover, the Chinese Pharmacopoeia also regulates Polygonatum cyrtonema Hua, which is also known as Huangjing. The sentence with“Polygonatum cyrtonema Hua is supervised by the Ministry of Health and Center for Food Evaluation( State Administration for Market Regulation) in China.” is added in the Introduction in corresponding lines 95-97.
Reviewer 2 Report
Comments and Suggestions for Authors
Manuscript title: The molecular mechanism of Polygonatum polysaccharide in improving hyperuricemia by regulating key targets of uric acid 2 metabolism in mice.
I have carefully evaluated your manuscript and commend your efforts in investigating the antihyperuricemia effect of PPCs extracted from PC, as well as the possible mechanism of action of PCPs. The topic is relevant and timely, given the growing interest in natural compounds and their therapeutic potential. However, several critical issues need to be addressed.
- Restructure the abstract section, describe briefly the methods used to evaluate the impact of PCPs on animal model and present the main obtained results.
- Scientific names should be italicized
- Introduction - Lacks Hypothesis and Defined Knowledge Gap
* While you provide a broad overview of kidney diseases and oxidative stress, there is no clearly defined knowledge gap or hypothesis.
* The introduction also repeats widely known facts without critical insight. - Dose Rationale and Toxicity
* The 125, 250 and 500 mg/kg doses are not justified based on prior safety studies, as well as there is no mention of potential toxicity. - provide kit codes used to determine all biochemical attributes
- Histopathological Analysis - Semi-Quantitative Scoring
* The histopathology was evaluated qualitatively, which limits reproducibility.
* A scoring system (e.g., 0-4 scale for inflammation, necrosis, etc.) was introduced only superficially.
Author Response
Comments 1: [Restructure the abstract section, describe briefly the methods used to evaluate the impact of PCPs on animal model and present the main obtained results.]
Response 1:
Thanks. According to your suggestion, we have rewritten the content of abstract as follows(line13-34): Polygonatum cyrtonema Hua, a plant with a long history of consumption in China, serves both medicinal and edible purposes, and it exhibits numerous pharmacological effects, including promoting kidney health and enhancing immune function. However, the effect and molecular mechanism of Polygonatum cyrtonema polysaccharides (PCPs) on hyperuricemia have not been reported yet. The hyperuricemic mice model was induced by intraperitoneal injection of potassium oxonate (PO, 300 mg/kg) combined with intragastric administration of hypoxanthine (HX, 300 mg/kg). Biochemical assays in mice revealed that PCPs markedly lowered high serum uric acid levels, suppressed xanthine oxidase (XOD) activity, and reduced the expression of inflammatory cytokines-including interleukin-1β(IL-1β), interleukin-6 (IL-6), and tumor necrosis factor-alpha (TNF-α). Western blot analysis demonstrated that PCPs down-regulated urate transporter 1 (URAT1), while H&E staining showed that PCPs effectively restored renal histological integrity. Here, we isolated and identified the PCPs, which consist mainly of rhamnose, glucuronic acid, galacturonic acid, glucose, galactose, and arabinose, with a molar mass ratio of 0.5:2.15:0.47:16.58:3.66:1.09. Further, the galactose residue docked with both XOD and URAT1 molecules forms more hydrogen bonds and exhibits lower binding energy, which enables improved regulation of both targets. We have demonstrated for the first time the improving effect of PCPs on hyperuricemia, and revealed their regulatory mechanisms by modulating xanthine oxidase, inflammatory factors and uric acid transporters. This study would not only provide the new insights into the anti-hyperuricemic activity of PCPs in mice, but also lay a foundation for its potential application in functional foods of anti-hyperuricemia.
Comments 2: [Scientific names should be italicized]
Response 2: Thanks for your suggestion. We replaced the sentences with “Polygonatum cyrtonema Hua (P. cyrtonema)”.
Comments 3:[ Introduction - Lacks Hypothesis and Defined Knowledge Gap
* While you provide a broad overview of kidney diseases and oxidative stress, there is no clearly defined knowledge gap or hypothesis.
* The introduction also repeats widely known facts without critical insight. ]
Response 3: Thanks for your suggestion. The investigation of PCPs in the Introduction is “P. cyrtonema polysaccharides (PCPs) demonstrate a variety of pharmacological effects, encompassing antioxidant properties, anti-aging, anti-microbial, enhancement of immune function, regulation of serum glucose and lipid levels, anticancer activities, obesity, protection of cardiocerebrovascular health, and anti-fatigue” in corresponding lines 100-104.
Comments 4:[ Dose Rationale and Toxicity
* The 125, 250 and 500 mg/kg doses are not justified based on prior safety studies, as well as there is no mention of potential toxicity.]
Response 4: Thanks for your suggestion. We referred to the dosage used in clinical practice of Huangjing and the dosage used as a common food in China, and then converted it to the dosage for mice. Polygonatum cyrtonema Hua is supervised by the Ministry of Health and Center for Food Evaluation (State Administration for Market Regulation) in China.
Comments 5:[ provide kit codes used to determine all biochemical attributes]
Response 5: Thanks for your suggestion. The kit codes are added as follows(line120-134):
1-phenyl-3-methyl-5-pyrazolone(PMP, P109105) was purchased from Shanghai Aladdin Biochemical Co., Ltd. (Shanghai,China).. Allopurinol(AP,Cat No:A800424), potassium oxonate (PO,Cat No:P831461) and hypoxanthine (HX,Cat No:H811076) were purchased from Shanghai Macklin Biochemical Co., Ltd. (Shanghai,China). Assay kits for determination of Serum UA(Cat No:105-000476-00), Urea (Cat No:105-000452-00)and creatinine (CREA, Cat No:105-000457-00) were obtained from Shenzhen Mindray Biomedical Electronics Co.,Ltd. (Shenzhen, China). Enzyme-linked immunoabsorbent commercial cytokine for murine TNF-α(Cat No:RX202412M), IL-1β(Cat No:RX203063M) and IL-6(Cat No:RX203049M) were bought from Quanzhou Ruixin Biotechnology Co., Ltd. (Quanzhou, China). Xanthine oxidase (XOD, Cat No:A002-1-1) assay kit was bought from Nanjing JianCheng Bioengineering Institute(Nanjing, China), Anti-URAT1(Cat No:14937-1-AP), anti-β-actin (Cat No:81115-1-RR) and secondary antibody (Cat No:SA00001-2) were purchased from Proteintech Group, Inc. (Wuhan, China), and Nitrocellulose Blotting Membrane(NC,0.45µm) was supplied by Amersham (Sheffield,UK ). Agilent 1260 HPLC was bought from Agilent Technologies, Inc.
Comments 6:[Histopathological Analysis - Semi-Quantitative Scoring
* The histopathology was evaluated qualitatively, which limits reproducibility.
* A scoring system (e.g., 0-4 scale for inflammation, necrosis, etc.) was introduced only superficially.]
Response 6: Thanks for your suggestion. The histopathology in the study is H&E, we can not use scoring system.
Reviewer 3 Report
Comments and Suggestions for Authors
In the present research, the authors performed the separation and purification of P. cyrtonema
polysaccharides and the identification of structural components. Furthermore, the molecular
docking simulation of polysaccharide residues and XOD and URAT1 target proteins was carried
out, while the efficacy of lowering serum uric acid of PCPs was examined in vivo. Although the
manuscript possesses scientific merit and provides well-established scientific results, some issues need to be addressed to improve the quality of the manuscript:
1) Please rephrase the confusing sentence within the Abstract section (Lines 26 and 27), as it
sounds a bit unclear. For example: “Further, the galactose residue docked with both XOD
and URAT1 molecules forms more hydrogen bonds and exhibits lower binding energy,
which enables improved regulation of both targets”.
2) In the Introduction section, please insert and present the chemical reactions of the uric acid
formation catalyzed by xanthine oxidase.
3) Introduction section (Line 59). Please specify the exact interleukins that are released within
the immune-inflammatory response.
4) Please specify the HPLC apparatus along with the manufacturer details.
5) Please provide a rationalization for the dosing of allopurinol. Considering that allopurinol
is administered orally, how is it dissolved, and in which solvent? Please provide details.
6) Please specify the structures of ligand molecules obtained from PubChem. Please provide
the PubChem CIDs for those molecules.
7) Please state the exact software versions of PyMOL and Discovery Studio visualizer.
8) Regarding the molecular docking protocol, is it semi-flexible docking? Please state this
explicitly. Also, is it focused or blind molecular docking? Indicate how the binding sites
on the target proteins were selected and prepared. Which scoring function of AutoDock
Vina was used? Default? State this clearly in text.
9) Figure 7, presenting the molecular docking results, is inappropriate. Within the 3D
interactions view, it is necessary to display molecular interactions in appropriate colors to
accurately present how they are formed. In the same Figure, it is also necessary to mark
and color the most important residues that interact with ligand atoms. On the other hand,
the Figures showing 2D interactions are of poor quality and are completely unreadable.
Please significantly improve their resolution and quality.
10) Specify the HPLC apparatus used, along with the manufacturer details.
11) The discussion regarding the obtained in silico results is completely missing. Please include
this in the Discussion section.
12) The obtained in silico results are not mentioned in the Conclusion section. Please mention
them in light of integration with the obtained experimental results.
13) At the end of the Conclusion, clearly state the main limitations of the current research.

Author Response
Dear editor and reviewers,
Thanks for your critical comments. According to your suggestions, we made revisions and response to reviewers one by one as follows. The comments were listed one by one in numerical order, followed by corresponding answers. The revisions in manuscript were highlighted in red.
Reviewer #3
Comments 1:[ Please rephrase the confusing sentence within the Abstract section (Lines 26 and 27), as it sounds a bit unclear. For example: “Further, the galactose residue docked with both XOD
and URAT1 molecules forms more hydrogen bonds and exhibits lower binding energy,
which enables improved regulation of both targets”.]
Response 1: Thank you for your revision of Lines 26 and 27. The sentence with “Further, the galactose residue docked with both XOD and URAT1 molecules forms more hydrogen bonds and exhibits lower binding energy, which enables improved regulation of both targets.” was added in the Abstract in lines 28-29.
Comments 2:[In the Introduction section, please insert and present the chemical reactions of the uric acid formation catalyzed by xanthine oxidase.]
Response 2: Thanks for your suggestion. Chemical reaction was added in the Introduction(line55-56): “Hypoxanthine + Hâ‚‚O + Oâ‚‚ → Xanthine + Hâ‚‚O₂,Xanthine +Hâ‚‚O + Oâ‚‚ → Uric Acid + Hâ‚‚Oâ‚‚”
Comments 3:[ Introduction section (Line 59). Please specify the exact interleukins that are released within the immune-inflammatory response.]
Response 3: Thanks for your suggestion. The sentence with “IL-1β and IL-6” was added in line 60.
Comments 4:[ Please specify the HPLC apparatus along with the manufacturer details.]
Response 4: Thanks for your suggestion. Agilent 1260 HPLC (Agilent Technologies, Inc. ) was added in Materials and Methods “line 134”.
Comments 5:[ Please provide a rationalization for the dosing of allopurinol. Considering that allopurinol is administered orally, how is it dissolved, and in which solvent? Please provide details.]
Response 5: Thanks for your suggestion. As for the rationalization for the dosing of allopurinol, we referred the literature “Effect of lemon water soluble extract on hyperuricemia in a mouse model” published on Food &Function in 2019.
As for solution, we performed the experiments as follows: 1. Prepare a 0.5mg/ml allopurinol solution using 0.5% sodium carboxymethyl cellulose (CMC-Na) as the solvent. 2. Detailed steps:
(1) First, prepare 0.5% CMC-Na. Weigh 0.50g of CMC-Na, add a small amount of deionized water, and stir slowly and thoroughly to allow for swelling and dissolution. Then, use deionized water to dilute to 100ml, thus obtaining a uniform and transparent solution of 0.5% CMC-Na.
(2) Preparation of 0.5mg/ml allopurinol solution: Weigh 0.0500g of allopurinol, first add a small amount of 0.5% CMC-Na and stir, then use 0.5% CMC-Na to dilute to 100ml.
Comments 6:[ Please specify the structures of ligand molecules obtained from PubChem. Please provide the PubChem CIDs for those molecules.]
Response 6: Thanks for your suggestion. Galactose, Compound CID: 6036; Galacturonic acid, Compound CID: 439215; Glucuronic acid, Compound CID: 94715; Arabinose, Compound CID: 439195. We added them between lines 234~235.
Comments 7:[ Please state the exact software versions of PyMOL and Discovery Studio visualizer.
Response 7: Thanks for your suggestion. “open-source PyMOL2.4.0, scovery Studio Visualizer 2025” was added in lines 243~244.
Comments 8:[ Regarding the molecular docking protocol, is it semi-flexible docking? Please state this explicitly. Also, is it focused or blind molecular docking? Indicate how the binding sites on the target proteins were selected and prepared. Which scoring function of AutoDock Vina was used? Default? State this clearly in text.]
Response 8: Thanks for your suggestion. We have rewritten molecular docking in the Method.The details are as follows(lines 239-251 ): The 3D structures of receptor proteins were downloaded from the PDB website, and those of ligand molecules were obtained from Pubchem. ligand molecules included Galactose(Compound CID: 6036), Galacturonic acid(Compound CID: 439215), Glucuronic acid(Compound CID: 94715), Arabinose(Compound CID: 43919). Using Autodock Vina, we performed molecular docking simulations between receptor proteins (like XOD ID:1FIQ and URAT1 ID:9JDV) and ligands (such as polysaccharide residues) to screen for binding interactions, calculate binding modes and affinities. Adopting a docking method where the protein is rigid while the ligand can rotate. The location of the receptor protein binding center is determined based on the ligand position from the PDB website. There is a fixed docking center location for URAT1 (center_x = 82, center_y = 94, center_z = 91) and XOD (center_x = 21, center_y = 16, center_z = 104). The scoring function utilized by AutoDock Vina was set to default. Visual analyses were performed by open-source PyMOL2.4.0 and Discovery Studio visualizer 2025.
Comments 9:[Figure 7, presenting the molecular docking results, is inappropriate. Within the 3D interactions view, it is necessary to display molecular interactions in appropriate colors to accurately present how they are formed. In the same Figure, it is also necessary to mark and color the most important residues that interact with ligand atoms. On the other hand, the Figures showing 2D interactions are of poor quality and are completely unreadable. Please significantly improve their resolution and quality.]
Response 9: Thanks for your suggestion. I revisualized the diagram of molecular docking (Figure 7 and 8).
Comments 10:[ 10) Specify the HPLC apparatus used, along with the manufacturer details.]
Response 10: Thanks for your suggestion. Agilent 1260 HPLC (Agilent Technologies, Inc.) was added in Materials and Methods “line 134”.
Comments 11:[ The discussion regarding the obtained in silico results is completely missing. Please include this in the Discussion section.]
Response 11: Thanks for your suggestion. The sentences with “Molecular docking studies reveal that PCPs, including galactose, galacturonic acid, and glucuronic acid, demonstrate lower binding energies with receptor proteins XOD and URAT1, primarily through hydrogen bonding interactions to regulate targets. In contrast, the interaction between arabinose and the URAT1 receptor is devoid of hydrogen bonds, indicating a relatively weaker binding affinity.” were added in the Discussion in corresponding lines 489-493.
Comments 12:[ The obtained in silico results are not mentioned in the Conclusion section. Please mention them in light of integration with the obtained experimental results.]
Response 12:
Thanks. The sentence with “Molecular docking studies suggest that resides of PCPs play a role in modulating two critical targets, XOD and URAT1 by hydrogen bonding interactions.” was added in the Conclusion section in corresponding lines 499-500.
Comments 13:[ At the end of the Conclusion, clearly state the main limitations of the current research.]
Response 13: Thanks. The sentence with “This study is subject to several limitations, primarily due to the reliance on a mice model without subsequent validation in rat models and human clinical trials.” was added in the Conclusion section in corresponding lines 504-506.
Reviewer 4 Report
Comments and Suggestions for Authors
Dear Authors,
Your work addresses a timely and relevant topic, and I appreciate the effort invested in this study. However, after a careful reading of your manuscript, I would like to offer the following detailed comments and suggestions that I believe are essential for improving the scientific rigor and clarity of your submission.
Below I summarize the main concerns:
Introduction
- The first mention of the plant should include the full botanical name with author citation, and all Latin names should be italicized.
- The introduction contains typographical errors (e.g., “bolongs”, “gouty”).
- The research objective is mentioned but remains vague; it should be clearly stated in one or two sentences.
- The rationale for investigating PCPs is not sufficiently explained.
Materials and Methods
- Several typographical errors are present (e.g., “kits was”).
- Units are incorrectly reported (e.g., membrane pore size “0.45 µM” instead of “0.45 µm”).
- Many key reagents lack company names and catalog numbers.
- The HPLC method is incomplete, with no detailed gradient composition or injection volume.
- No information is provided on ethical approval for animal experiments.
- Histology lacks scoring methodology and details of sample preparation.
- Statistical analysis is insufficient; one-way ANOVA with post hoc testing should be used instead of multiple t-tests.
Results and Discussion
- The statement that PCPs showed “comparable effects to allopurinol” is an overinterpretation. Allopurinol is a clinically validated reference drug, whereas this study is limited to a murine model.
- The novelty of the findings should be emphasized more clearly, particularly how PCPs compare with other natural polysaccharides (e.g., Lonicera, Enteromorpha, corn silk).
- The discussion is overly long and contains excessive general background on polysaccharides, which detracts from the focus on the study’s own results.
Conclusion
- The limitations of the study (e.g., animal model, lack of clinical validation) should be explicitly acknowledged.
- The authors should highlight that this is the first study to demonstrate the hypouricemic potential of PCPs.
References
- Reference formatting is inconsistent with Foods (MDPI) style.
- Journal abbreviations are not standardized according to ISO-4.
- Latin names in titles should be italicized.
- Recent systematic reviews and clinical studies on hyperuricemia (2022–2024) are missing.
- DOIs should be added wherever available.
In my comments
Author Response
Dear editor and reviewers,
Thanks for your critical comments. According to your suggestions, we made revisions and response to reviewers one by one as follows. The comments were listed one by one in numerical order, followed by corresponding answers. The revisions in manuscript were highlighted in red.
Reviewer #4
Comments 1:[Introduction
The first mention of the plant should include the full botanical name with author citation, and all Latin names should be italicized.]
The introduction contains typographical errors (e.g., “bolongs”, “gouty”).
The research objective is mentioned but remains vague; it should be clearly stated in one or two sentences.
The rationale for investigating PCPs is not sufficiently explained.
Response 1: Thanks for your suggestion. Our modifications are as follows: We replaced the sentences with “Polygonatum cyrtonema Hua (P. cyrtonema)” “bolongs” is replaced with “belongs” in line 83. The investigation of PCPs is “P. cyrtonema polysaccharides (PCPs) demonstrate a variety of pharmacological effects, encompassing antioxidant properties, anti-aging, anti-microbial, enhancement of immune function, regulation of serum glucose and lipid levels, anticancer activities, obesity, protection of cardiocerebrovascular health, and anti-fatigue” in corresponding lines 100-104.
Comments 2:[Materials and Methods
Several typographical errors are present (e.g., “kits was”).
Units are incorrectly reported (e.g., membrane pore size “0.45 µM” instead of “0.45 µm”).
Many key reagents lack company names and catalog numbers.
The HPLC method is incomplete, with no detailed gradient composition or injection volume.
No information is provided on ethical approval for animal experiments.
Histology lacks scoring methodology and details of sample preparation.
Statistical analysis is insufficient; one-way ANOVA with post hoc testing should be used instead of multiple t-tests.
Response 2: Thanks for your suggestion. Our modifications are as follows: “kits was” is replaced with “ kits were”; 0.45 µM” is replaced with “0.45 µm”. Company names and catalog numbers were added in corresponding lines 120-134. The sentence with “The mobile phase was an acetonitrile: (pH = 7.0) phosphate buffer=16%:84% gradient elution at a flow rate of 1 mL/min.” is added in corresponding lines 171-172. The sentence with “HPLC injection volume was 20µL.” was added in corresponding lines 171-172. We provided ethical approval for animal experiments to the editor of journal-foods. The histopathology in the study is H&E, and hence, we can not use scoring system.
The detailed method of (H&E) Paraffin section making was described as follows:
- Tissue fixation: Fresh tissue is immediately put into tissue fixative for more than 24 hours, stored and transported at room temperature. Take the tissue out of the fixative and use the Scalpel to smooth the target tissue in the fume hood. Put the cut tissue and the corresponding label in the embedding frame.
- Dehydration and wax leaching: put the dehydration box into the dehydrator in order to dehydrate with gradient alcohol. 75% alcohol for 4 hours,85% alcohol for 2 hours,90% alcohol for 2 hours,95% alcohol for 1 hour,anhydrous ethanol I for 30 min,anhydrous ethanol II for 30 min, alcohol benzene for 5~10 min,xylene II for 5~10 min,65℃melting paraffin I for 1h,65 ℃ melting paraffin II for 1h,65℃ melting paraffin III for 1 hour.
- Embedding: The wax-soaked tissue is embedded in the embedding machine. First, put the melted wax into the embedding frame, and before the wax solidifies, remove the tissue from the dewatering box and put it into the embedding frame according to the requirements of the embedding surface and affix the corresponding label. Cool at-20 °frozen platform, and after the wax is solidified, the wax block is removed from the embedding frame and repaired.
- Section: Put the trimmed wax block into a paraffin slicer for slicing, with a thickness of 4 μ M. The tissue is flattened when the slice floats on the 40 ℃ warm water of the spreading machine, and the tissue is picked up by the glass slides and baked in the oven at 60 ℃. After the water-baked dried wax is melted, it is taken out and stored at room temperature.
Regarding the use of statistical methods, we mainly conduct t-tests using models as controls, referring the similar research as follows:
Yongmei Li, Zean Zhao, Jian Luo, et al, Apigenin ameliorates hyperuricemic nephropathy by inhibiting URAT1 and GLUT9 and relieving renal fibrosis via the Wnt/β-catenin pathway, Phytomedicine,Volume 87,2021,153585. https://doi.org/10.1016/j.phymed.2021.153585.
Comments 3:[Results and Discussion
The statement that PCPs showed “comparable effects to allopurinol” is an overinterpretation. Allopurinol is a clinically validated reference drug, whereas this study is limited to a murine model.
The novelty of the findings should be emphasized more clearly, particularly how PCPs compare with other natural polysaccharides (e.g., Lonicera, Enteromorpha, corn silk).
The discussion is overly long and contains excessive general background on polysaccharides, which detracts from the focus on the study’s own results.]
Response 3: Thanks for your suggestion. Our modifications are as follows: We limited the comparison to only “HUA mice” in corresponding line 462-463. The sentence with “PCPs demonstrate efficacy in lowering uric acid levels by inhibiting XOD activity, downregulating urate transporter 1 (URAT1), and exhibiting anti-inflammatory effects, and polysaccharides from Lonicera japonica have similar efficacy.” was added in the discussion in corresponding lines 486-489. We compress the background on polysaccharides in the discussion in corresponding lines 420-432.
Comments 4:[Conclusion
The limitations of the study (e.g., animal model, lack of clinical validation) should be explicitly acknowledged.
The authors should highlight that this is the first study to demonstrate the hypouricemic potential of PCPs.
Response 4: Thanks for your suggestion. Our modifications are as follows: The sentence with “This study is subject to several limitations, primarily due to the reliance on a mice model without subsequent validation in rat models and human clinical trials.” was added in conclusion in corresponding lines 504-506. The sentence with “The present study demonstrates that PCPs firstly reduce serum uric acid in mice with hyperuricemia.” was added in conclusion in corresponding line 495-496.
Comments 5:[References
Reference formatting is inconsistent with Foods (MDPI) style.
Journal abbreviations are not standardized according to ISO-4.
Latin names in titles should be italicized.
Recent systematic reviews and clinical studies on hyperuricemia (2022–2024) are missing.
DOIs should be added wherever available.
Response 4:Thanks for your suggestion. Our modifications are as follows: Foods (MDPI) style is revised. Latin names “P. cyrtonema” is italicized with “P. cyrtonema”.
A new citation of “Asghari, K.M.; Zahmatyar, M.; Seyedi, F.; Motamedi, A.; Zolfi, M.; Alamdary, S.J.; Fazlollahi, A.; Shamekh, A.; Mousavi, S.E.; Nejadghaderi, S.A.; et al. Gout: global epidemiology, risk factors, comorbidities and complications: a narrative review. BMC Musculoskelet. Disord. 2024, 25, 1047.” was added in the Introduction section.
DOIs were added in the Reference section.
Reviewer 5 Report
Comments and Suggestions for Authors
The study is well-investigated but requires some further clarification in certain areas. The figures should be self-explanatory to the readers. The description can be improved for better clarity. A visual representation of the dosing can improve understanding of the experimental setup.

The manuscript requires thorough proofreading, as there are grammatical errors throughout the paper. I will suggest carefully reviewing the text and addressing them.
Author Response
Dear editor and reviewers,
Thanks for your critical comments. According to your suggestions, we made revisions and response to reviewers one by one as follows. The comments were listed one by one in numerical order, followed by corresponding answers. The revisions in manuscript were highlighted in red.
Reviewer #5
- Scientific names
- Ensure that all scientific names are correctly formatted. For example, P. cyrtonema should be italicized. Please review the manuscript for similar instances.
Response 1:Thanks for your suggestion. Our modifications are as follows: P. cyrtonema is italicized with “P. cyrtonema”.
- Abbreviations:
- Expand all abbreviations upon their first time mentioning them before using their abbreviated form. Example in line 152, NaOH should be written as Sodium hydroxide (NaOH). Please proofread for such inconsistencies.
Response 2: Thanks for your suggestion. Our modifications are as follows: NaOH is replaced with
“Sodium hydroxide (NaOH)” in line 161.
- Introduction:
- Include the citation for the following study in the introduction: “Mao, H., Huang, S., Lin, T., Ding, Y., & Yang, Z. (2025). Exploring the relationship between circadian syndrome, serum uric acid levels, and hyperuricemia: evidence from NHANES 2005–2018. Scientific Reports, 15(1), 28984.” In the introduction
- Line 50-52 contain grammatical errors and should be rewritten for clarity.
- Line 59, there is mention of “IL” without clarification. Please clarify which interleukin it is being referred to.
Response 3:Thanks for your suggestion. Our modifications are as follows:
a.“Mao, H., Huang, S., Lin, T., Ding, Y., & Yang, Z. (2025). Exploring the relationship between circadian syndrome, serum uric acid levels, and hyperuricemia: evidence from NHANES 2005–2018. Scientific Reports, 15(1), 28984.” is citated in the introduction
- Line 50-52 : New sentences( lines 50-52) “The prevalence of hyperuricemia in China has been rising annually, with a notably higher incidence observed among men compared to women. The age of onset among men is showing a younger trend.”
- Line 59, “ IL-1β and IL-6 “ is added in line 60
- Material and methods a. Include a figure or a table for the dosing of each group. The current description is unclear and visual representation will help the readers.
Response 4: Thanks for your suggestion. we make table 1 for the dosing of each group in the Material and methods in lines 188.
Table 1 The dosing of each group in mice
|
Group |
Treatment |
|
|
NC |
0.5% CMC-Na+ 0.5% CMC-Na |
0.5% CMC-Na |
|
MC |
PO(300mg/kg/d)+HX(300mg/kg/d) |
0.5% CMC-Na |
|
AP |
PO(300mg/kg/d)+HX(300mg/kg/d) |
AP(300mg/kg/d) |
|
LD |
PO(300mg/kg/d)+HX(300mg/kg/d) |
PCPs(125mg/kg/d) |
|
MD |
PO(300mg/kg/d)+HX(300mg/kg/d) |
PCPs(250mg/kg/d) |
|
HD |
PO(300mg/kg/d)+HX(300mg/kg/d) |
PCPs(500mg/kg/d) |
normal control group (NC), model control group (MC), allopurinol(AP), PCPs- low dose group (LD), PCPs-medium dose group (MD) and PCPs-high dose group (HD).
- Figures
- Revise the figure legend to include details such mice treatment group, information of dosing, and definition of each acronym used.
- Figure 5: the legend mentions presence of arrows; however, no arrows are visible in the figure. Please correct this.
- Figure 6: housekeeping protein used is b-actin, I will recommend on adding beta before actin
- Figure 7: It is not legible. Please increase the font size or adjust the layout to improve visibility
Response 5:Thanks for your suggestion. Our modifications are as follows:
a. All figure legends were added with “normal control group (NC), model control group (MC), allopurinol (AP), PCP- low dose group (LD), PCP-medium dose group (MD) and PCP-high dose group (HD).”.
b.c.d: Figure 5,6,7 are replaced with new Figures which are clear.
Round 2
Reviewer 2 Report
Comments and Suggestions for Authors
All issues have been raised.
Reviewer 3 Report
Comments and Suggestions for Authors
The authors have conducted a significant revision of the manuscript, significantly improving its quality, making it now suitable for publication. Good work.
Reviewer 4 Report
Comments and Suggestions for Authors
All remarks have been adedd